# Design of a Multi-Epitope Vaccine against Tuberculosis from *Mycobacterium tuberculosis* PE_PGRS49 and PE_PGRS56 Proteins by Reverse Vaccinology

**DOI:** 10.3390/microorganisms11071647

**Published:** 2023-06-24

**Authors:** Maritriny Ruaro-Moreno, Gloria Paulina Monterrubio-López, Abraham Reyes-Gastellou, Juan Arturo Castelán-Vega, Alicia Jiménez-Alberto, Gerardo Aparicio-Ozores, Karen Delgadillo-Gutiérrez, Jorge Alberto González-Y-Merchand, Rosa María Ribas-Aparicio

**Affiliations:** 1Departamento de Microbiología, Escuela Nacional de Ciencias Biológicas, Instituto Politécnico Nacional (IPN), Mexico City 11340, Mexico; mruaro4@gmail.com (M.R.-M.); gmonterrubiol@ipn.mx (G.P.M.-L.); summoncerberus@hotmail.com (A.R.-G.); gaparico@ipn.mx (G.A.-O.); kdelgadillog@ipn.mx (K.D.-G.); 2Posgrado en Biomedicina y Biotecnología Molecular, Escuela Nacional de Ciencias Biológicas, Instituto Politécnico Nacional (IPN), Mexico City 11340, Mexico; jgonzal1212@yahoo.com.mx

**Keywords:** B- and T-cell epitope prediction, tuberculosis, vaccine, computational vaccinology

## Abstract

Tuberculosis is a disease caused by *Mycobacterium tuberculosis*, representing the second leading cause of death by an infectious agent worldwide. The available vaccine against this disease has insufficient coverage and variable efficacy, accounting for a high number of cases worldwide. In fact, an estimated third of the world’s population has a latent infection. Therefore, developing new vaccines is crucial to preventing it. In this study, the highly antigenic PE_PGRS49 and PE_PGRS56 proteins were analyzed. These proteins were used for predicting T- and B-cell epitopes and for human leukocyte antigen (HLA) protein binding efficiency. Epitopes GGAGGNGSLSS, FAGAGGQGGLGG, GIGGGTQSATGLG (PE_PGRS49), and GTGWNGGKGDTG (PE_PGRS56) were selected based on their best physicochemical, antigenic, non-allergenic, and non-toxic properties and coupled to HLA I and HLA II structures for in silico assays. A construct with an adjuvant (RS09) plus each epitope joined by GPGPG linkers was designed, and the stability of the HLA-coupled construct was further evaluated by molecular dynamics simulations. Although experimental and in vivo studies are still necessary to ensure its protective effect against the disease, this study shows that the vaccine construct is dynamically stable and potentially effective against tuberculosis.

## 1. Introduction

The World Health Organization (WHO) estimated that in 2021, 10.6 million people acquired tuberculosis (TB), an increase of 4.5% over cases in 2020, and 1.6 million people died of TB [1]. TB is a reemerging disease because of strain selection due to antimicrobial resistance and coinfections with HIV [2,3]. Therefore, the development of a new TB vaccine is essential.

The TB etiologic agent is Mycobacterium tuberculosis (MTB), which belongs to the Mycobacterium tuberculosis complex (MTBC) or clade “Tuberculosis-Simiae” [4,5]. It is an acid-fast bacterium, and its genome comprises approximately 4.4 Mbp. A striking feature of the MTB genome is the presence of PE and PPE family genes [6]. The PE and PPE protein families are unique and plentiful in mycobacteria, occupying almost 10% of the coding capacity of the MTB genome [7,8]. PE protein subgroups can be identified based on their sequence phylogeny and classified into five sublineages (PE35, PE5, PE36/PE25, ESX-5 secreted proteins, and PGRS) [9]. The PGRS subgroup consists of GC-rich polymorphic sequences available in multiple copies on surface-exposed proteins, restricted to the MTBC and some other pathogenic mycobacteria. There is evidence that their antigenic variability facilitates the uptake of mycobacteria by macrophages, but little is known about their function after MTB is phagocytosed by host macrophages [8,10,11,12]. Some studies reported that the PE_PGRS subfamilies have the potential to generate a strong immune response within the host cell. Experimentally, it was observed that the immunization of mice with the PE domain induces a cellular immune response [13,14,15] and, therefore, could be a good vaccine candidate.

Reverse vaccinology approaches use computational methods and tools to identify antigens and epitopes that are likely to be recognized by the immune system and induce a protective response. Compared to in vitro studies, reverse vaccinology is performed in less time, saving economic resources and experimental work to support successful vaccine development [16,17].

In a previous study by our group, proteins PE-PGRS49 and PE-PGRS56 from the MTB H37Rv proteome were identified as highly antigenic molecules [18]. PE-PGRS proteins contain an N-terminal Pro-Glu domain and a domain comprised of repetitive sequences such as GGAGGX, where X can be any amino acid [19]. PE-PGRS are present in different mycobacterial species [20], are produced by mycobacteria during infection, and could function as adhesion factors [21]. Accordingly, this work aimed to in silico study these particular proteins and their antigenic regions to select epitopes that could be used, after appropriate experimental and in vivo studies, in a new vaccine against TB.

## 2. Materials and Methods

### 2.1. Amino Acid Sequences of PE_PGRS49 and PE_PGRS56

The amino acid sequences of the PE-PGRS49 and PE-PGRS56 proteins from MTB H37Rv were downloaded in FASTA format from the National Center for Biotechnology Information (NCBI, https://www.ncbi.nlm.nih.gov/ (accessed on 1 February 2023) with reference numbers YP_177961.2 and Rv3512, respectively.

### 2.2. Prediction of B-Cell Epitopes

The Immunomedicine tools server (http://imed.med.ucm.es/Tools/antigenic.pl (accessed on 1 February 2023) was used to predict B-cell epitopes. This program is developed to predict antigenic determinants and it is based on a semi-empirical method which use physicochemical properties of amino acid residues and experimental data. The antigenic peptides are determined by the Kolaskar and Tongaonkar method [22].

### 2.3. Selection of Frequent Alleles in the World’s Population

The Allele Frequency Net Database [23] (http://allelefrequencies.net/ (accessed on 6 February 2023) was employed for the prediction of MHC epitopes, where frequencies of alleles, haplotypes, and genotypes of different world populations are selected to choose the most frequent alleles in the population [24], selecting those that cover more than 95% of the world’s population [25].

### 2.4. Prediction of Epitopes of T Cells (HLA Classes I and II)

The servers used to predict peptide binding to HLA Class I and II molecules were Propred1, which is an online service for identifying the MHC Class-I binding regions in antigen. ProPred1 also allows the prediction of the standard proteasome and immunoproteasome cleavage sites in an antigenic sequence. Thus, the prediction of MHC binders and proteasome cleavage sites in an antigenic sequence leads to the identification of potential T-cell epitopes (https://webs.iiitd.edu.in/raghava/propred1/index.html (accessed on 14 February 2023) [26], RANKPEP allows the prediction of peptide binding to MHC-I and MHC-II molecules using motif profiles and a greater specificity of CD8 T-cell and CD4 T-cell epitopes (http://imed.med.ucm.es/Tools/rankpep.html (accessed on 15 February 2023) [27], and IEDB (Immune Epitope Database and Analysis Resource server) is an online server that is employed for the prediction of T cells. This server predicts epitopes based on receptor affinity (http://tools.iedb.org/mhcii/ (accessed on 16 February 2023) [28]. A window of nine amino acids per epitope was selected, and the default criteria were used.

### 2.5. Prediction of Antigenicity

The antigenicity of the epitopes obtained previously was predicted by the VaxiJen v2.0 server (http://www.ddg-pharmfac.net/vaxijen/VaxiJen/VaxiJen.html (accessed on 22 February 2023) using the default values [29]. The VaxiJen server ranks peptides based on the physical and chemical properties of amino acids with an alignment-independent auto-covariance and cross-covariance transformation. The threshold for antigenicity was set at 0.4.

### 2.6. Characterization of Selected B- and T-Cell Epitopes

All selected epitopes were analyzed by predicting various physical and chemical parameters. The molecular weight, theoretical pI, amino acid composition, atomic composition, estimated half-life time, and instability index were estimated with the ExPASy server (https://web.expasy.org/protparam/ (accessed on 22 February 2023) [30]. The allergenicity of the B- and T-cell epitopes was evaluated by AllergenFP v.1.0 (http://ddg-pharmfac.net/AllergenFP/ (accessed on 23 February 2023) [31], based on amino acid properties like hydrophobicity, size, relative abundance, helix, and β-strand-forming propensities. Epitope toxicity was evaluated with the ToxinPred server (https://webs.iiitd.edu.in/raghava/toxinpred/index.html (accessed on 23 February 2023). The server uses amino acid composition-based models, considering that some patterns/motifs could be associated with peptide toxicity [32].

### 2.7. Prediction of Docked Epitopes to MHC Protein

For molecular docking of epitopes, the crystal structure of molecules in complex with their self-peptides was obtained from the PDB (Protein Data Bank) database (https://www.rcsb.org (accessed on 24 February 2023). Proteins were prepared before docking by removing water molecules and each crystallized ligand with PyMOL (https://www.charmm.org/ (accessed on 27 February 2023). Additionally, energy minimization was performed on each structure. The epitopes were docked with the following MHC alleles: HLA-A0101 (PDB ID: 3BO8), HLA-A0201 (PDB ID: 4UQ3), HLA-A0301 (PDB ID: 2XPG), HLA-A2402 (PDB ID: 5HGH), HLA-B5801 (PDB ID: 5IM7), DRB1-0401 (PDB ID: 2SEB), and DRB1-1501 (PDB ID: 1BX2). Peptide-protein docking was performed using the HPEPDOCK server (http://huanglab.phys.hust.edu.cn/hpepdock/ (accessed on 1 March 2023) The server implements a hierarchical docking protocol with fast conformational sampling of peptide conformations and ensemble docking of generated peptide conformations against the protein. The docking server accepts both sequence and structure as input for proteins/peptides [33] for both HLA-I and HLA-II, using default server parameters and targeting the peptide binding cleft. The PyMOL tool (http://pymol.org (accessed on 6 March 2023) was used to visualize and analyze docked complexes.

### 2.8. Conservation of Selected Epitopes

A total of 1229 E-PGRS49 and 161 PE-PEPGRS56 sequences were downloaded from the NCBI database and aligned with MAFFT [34]. Aligned sequences were submitted to the WebLogo server [35] to obtain a visually simplified representation of the amino acid conservation in the alignment. This graphical logo was used to determine the conservation of the selected epitopes.

### 2.9. Construction of Vaccine Candidate

The epitopes were selected based on the best antigenicity value, non-allergenic and non-toxic prediction, and the property of forming epitope-protein complexes with the lowest global energy (most negative) in the molecular docking assays. These epitopes were thus inferred to be antigenic given the prediction results, and therefore used to construct the vaccine candidate, with the linker GPGPG connecting them. Additionally, the adjuvant RS09, which is a short peptide (APPHALS) that mimics bacterial lipopolysaccharide [36], was connected to the epitopes by the EAAAK linker [37].

### 2.10. Vaccine Candidate Molecular Docking with an Immune Receptor (TLR-4)

For molecular docking of the vaccine candidate, the crystal structure of TLR-4 (PDB ID: 4G8A) was obtained from the PDB (Protein Data Bank) database (https://www.rcsb.org (accessed on 5 April 2023). Protein was prepared before docking by removing water molecules and each crystallized ligand with PyMOL (https://www.charmm.org/ (accessed on 7 April 2023). Additionally, energy minimization was performed on the structure. The vaccine candidate was docked using the HDOCK server (http://hdock.phys.hust.edu.cn/ (accessed on 9 April 2023) [38] using default server parameters. The PyMOL tool (http://pymol.org (accessed on 25 April 2023) was used to visualize and analyze docked complexes.

### 2.11. Molecular Dynamics Simulation

The stability of peptides bound to MHC proteins was assessed by molecular dynamics simulations. Peptide structures were predicted with HPEPDOCK and then docked into the MHC I and II structures. The protein-peptide complexes were prepared with the Charmm-GUI solution builder, and each system was solvated using the TIP3P water model and neutralized with KCl at a final concentration of 0.15 M. Then, these structures were employed for molecular dynamics simulation performed with Gromacs v2018.3 and the CHARMM36m force field. Molecular dynamics simulation was performed at 310 K in three stages: 5000 steps of energy minimization, 125,000 steps of equilibration (NVT ensemble), and 50 ns of molecular dynamics production (NPT ensemble). Trajectory analysis was performed with Gromacs, and plots were generated with Grace 5.1.25 to visualize the stability of bound epitopes or vaccine candidates to MHC proteins, respectively.

## 3. Results

### 3.1. Prediction of B-Cell Epitopes

Twelve linear epitopes with a range from 7 to 15 amino acids were identified in the PE_PGRS49 protein using the IEDB server, while in the PE-PGRS56 protein, 33 linear epitopes with a range from 7 to 19 amino acids were identified. Epitopes predicted in both proteins are shown in the Appendix A.

### 3.2. Prediction of T-Cell Epitopes

For MHC Class I, alleles HLA-A0101, HLA-A0201, HLA-A0301, HLA-A2402, HLA-B0702, HLA-B0801, HLA-B1501, HLA-B5801, and HLA-B3901 were selected as representatives in the population, and for the prediction of MHC Class II epitopes, DRB1*0101, DRB1*0301, DRB1*0401, DRB1*0701, DRB1*0801, DRB1*1101, DRB1*1301, and DRB1*1501 alleles were chosen.

A total of 16 MHC Class I linear epitopes were predicted for the PE_PGRS49 protein by using the ProPred I server and one epitope with the Rankpep server. The amino acid residues that were most frequently recognized were located at positions 61–73, 110–124, 128–136, 222–233, and 274–286. In the PE_PGRS56 protein, we identified one epitope with ProPred I in the region 832–839; however, no epitopes were predicted with Rankpep.

Regarding the prediction of MHC Class II epitopes for the PE_PGRS49 protein, 10 epitopes were identified with the Rankpep server. The regions most frequently recognized were 177–187, 222–233, 262–271, and 274–286. On the other hand, 26 epitopes were predicted in the PE_PGRS56 protein. The regions most frequently recognized were 102–112, 338–347, 579–587, 780–789, and 1034–1044. The allele DRB1*0401 showed more matches on the RANKPEP and IEDB servers. All epitopes predicted in both proteins are shown in the Appendix A.

### 3.3. Selection of Epitopes

Epitopes that were predicted to be recognized by both T and B cells were the first selection criteria. Six epitopes of the PE_PGRS49 protein and five of the PE_PGRS56 protein were thus selected (Table 1).

### 3.4. Prediction of Epitope Antigenicity

The prediction of peptide antigenicity was performed with the VaxiJen server; the higher the value calculated with the server, the higher the antigenicity level (a score greater than 0.4 is considered antigenic by the VaxiJen server). For the PE_PGRS49 protein, the scores ranged from 0.0638 to 2.2969, of which peptide 177–187 does not have antigenic qualities according to the prediction. Similarly, in the prediction of peptide antigenicity for the PE_PGRS56 protein, scores ranged from 0.9916 to 2.2536, all having antigenic qualities (Table 1).

### 3.5. Characterization of Selected B- and T-Cell Epitopes

Some chemical and physical properties were investigated with the Expasy server: molecular weight, instability index, theoretical pI, and half-life time. Peptides predicted in PE_PGRS49 had a molecular weight between 919.90 and 1365.46, with a pI in the range of 5.52–8.47. The half-life time estimated in hours ranged from 1.9 h to 30 h. In addition, almost all peptides were predicted to be stable in vitro, according to the predictor. Predicted epitopes in PE_PGRS56 had a range of 10 to 15 amino acids, their molecular weight was 702.72–1201.26, the isoelectric point ranged between 5.19–5.84, and the estimated half-life time ranged from 1.1 h to 30 h. One epitope in PE_PGRS56 was predicted to be unstable. None of the epitopes were toxic, and most were non-allergenic, according to the ToxinPred and AllergenFP servers, respectively (Table 1).

### 3.6. Docking of Epitopes to MHC Class I and II Proteins

We performed molecular docking to ascertain if the MHC binding cleft recognizes the epitopes. The selected epitopes were docked with some of the most frequent MHC Class I and II alleles in the population, HLA*A0101, HLA*A0201, HLAA*0301, HLA*A2401, HLA*B5801, DRB1*0401, and DRB1*1501, using an online server, HPEPDOCK. Ten complexes were generated, and the most suitable epitope–MHC allele complex was selected based on correct conformation, binding, and the lowest energy level. Each docked epitope showed different energy values with their respective HLA alleles; the peptides 58–73, 128–138, 222–233, and 274–286 from the PE-PGRS49 protein scored the best binding energies and conformations. Although the 177–187 epitope showed affinity and low energy levels in complex with all HLA alleles, it was discarded because it was not identified as antigenic. In the case of the PE-PGRS56 protein, the epitopes with the best bound conformations were 579–593, 780–789, and 1034–1045 (Figure 1, Table 2).

### 3.7. Molecular Dynamics Simulations of Peptide Complexes

Molecular dynamics (MD) simulations were performed to verify the stability of predicted epitopes complexed with MHC II proteins (DRB1*0401 and DRB1*1501 alleles) and for the construction of the vaccine candidate with the same MHC proteins (Appendix A). The root mean square deviation (RMSD) plot showed the stability of the peptides 222–233 from PE_PGRS49 and 1034–1045 from PE_PGRS56 (Figure 2A). The stability of the complex was observed in the plot, which indicates that the protein and the peptide have similar behavior; therefore, their union is maintained.

### 3.8. Vaccine Candidate

Epitopes that were antigenic, non-allergic, non-toxic, and had a high binding affinity and stability with multiple MHC alleles were selected for the design of the vaccine candidate. Moreover, conservation analysis showed that all selected epitopes are conserved among all *M. tuberculosis*-reported proteins (Appendix A). Therefore, a linear vaccine sequence comprising three epitopes from PE_PGRS49 and one from PE_PGRS56 was constructed: GGAGGNGSLSS, FAGAGGQGGLGG, GIGGGTQSATGLG (PE_PGRS49), and GTGWNGGKGDTG (PE_PGRS56) (Figure 3). GPGPG linkers were used to join epitopes to prevent junctional epitope formation. Additionally, an LPS-like adjuvant (RS09) was attached at the amino-terminal end with the linker sequence EAAAK. The sequence of the vaccine candidate consists of 81 amino acids and has a molecular weight of 6600.97, according to prediction with the Expasy server.

### 3.9. Vaccine Candidate Molecular Docking with an Immune Receptor (TLR-4)

The vaccine candidate was docked into TLR-4 using the server HDOCK. Ten complexes were generated, and the most suitable complex, based on correct conformation and binding, was selected. An interaction was observed between the adjuvant, the epitopes, and TLR-4 with a docking score of −262.57 kcal/mol (Figure 4E).

### 3.10. Molecular Dynamic Prediction for the Vaccine Candidate

Molecular docking showed that the vaccine construct binds to the pocket of HLA proteins, showing low values of interaction energy (Figure 4A,B). In the case of the vaccine candidate (selected epitope construct) and MHC proteins, the RMSD plot also showed that the interaction between the vaccine candidate and the MHC molecules acquired a stable and compact form during the 50 ns of molecular dynamics simulation (Figure 2B). This stable binding suggests that the vaccine candidate could be recognized by proteins of the immune system and induce an adequate immune response.

## 4. Discussion

Developing a vaccine is crucial to eradicating or preventing the spread of infectious diseases such as MTB. An epitope vaccine could avoid false positives in TB diagnostic tests, as is the case with the BCG vaccine [39], and the pathogen has no virulence reversal risk. This study aimed to develop a vaccine candidate based on epitopes of two proteins from the MTB PE_PGRS subfamily that could elicit a protective immune response against TB. Bioinformatic tools have been used to develop epitope vaccines [40,41], and vaccines that elicit a T-cell response are already being developed against TB [42,43,44,45].

There are bioinformatics tools that facilitate the identification of vaccine candidates using less time and resources than classical vaccinology. They allow the identification of proteins and antigenic epitopes in B cells to find those responsible for stimulating both humoral and cell-mediated immunity [42,43]. The most efficient way to control diseases such as TB and many intracellular infections is through the identification of T-cell peptides that interact with the peptide-HLA complex and that can cause a cellular immune response [46,47]. In a previous study, the PE_PGRS49 and PE_PGRS56 proteins were identified as highly antigenic in the MTB proteome; consequently, their epitopes were identified and analyzed [18]. In our work, the regions of the protein sequence recognized by T and B cells were identified. A combination of multiple servers was used with different algorithms to obtain a more accurate selection of peptides. For this purpose, epitopes that bind to different HLA I and HLA II molecules were identified. Epitope prediction analysis of the PE_PGRS49 protein revealed that FAGAGGQGGLGG (222–233), SGFFGGKGGFG (177–187), and GIGGGTQSATGLG (274–286) were shown to have a higher affinity for B cells and HLA proteins. Epitope prediction analysis of the PE_PGRS56 protein revealed that DAGKAGTGSAPGT (101–113), TVGGGTVPAGSGGQG (579–593), and GTGWNGGKGDTG (1034–1045) demonstrated a higher affinity for B cells and HLA proteins. From the protein PE_PGRS49, the epitopes selected were those in positions 128–138, 222–233, and 274–286, and from the protein PE_PGRS56, the region of the amino acid sequence of the selected protein was 1034–1045. The filters considered were epitopes that can be recognized by T and B cells, antigenicity value, stability in docking, and molecular dynamics with HLA molecules, as well as being non-allergenic and non-toxic. The purpose of selecting several epitopes was to obtain the ability to simultaneously activate humoral and cellular immune responses, triggering a solid and enduring immune response [28,42].

The antigenicity value of each epitope was obtained in silico with the VaxiJen v.2.0 server, which allows for the classification of antigens based on physicochemical properties [29]. Priority was given to epitopes in the analysis with an antigenicity value greater than 1. The epitope SGFFGGKGGFG (177–187) of the PE_PGRS49 protein was discarded as it was not antigenic. Fortunately, in the case of the PE_PGRS56 protein, all the epitopes had antigenic quality. The selected epitopes had a much higher antigenicity value than other epitopes used for developing TB epitope vaccines and did not exceed values greater than 1.5 using the same server [42,43,44]. The more antigenic epitopes were FAGAGGQGGLGG (222–233), GIGGGTQSATGLG (274–286), and GTGWNGGKGDTG (1034–1045).

Bioinformatics tools were used to measure and evaluate physicochemical, antigenic, allergenic, and toxic parameters for designing vaccines with epitopes [29,30,31,32]. To be considered safe, an epitope vaccine cannot cause a hypersensitive reaction or toxicity within the host [31,48]. The allergenicity and toxicity of each epitope were determined, and those epitopes that were probable allergens or toxic were rejected. The physicochemical properties of epitopes were also calculated to determine each peptide’s nature, molecular weight, theoretical IP, stability, and estimated half-life. Selected epitopes were classified as stable.

T cells of the adaptive immune system are an essential mechanism for host recognition and control of TB infection. This recognition of antigens by T cells depends on binding epitopes to the HLA proteins [49]. The epitopes were docked with the following alleles from HLA I and II proteins: HLAA*0101, HLAA*0201, HLAA*0301, HLAA*0401, HLAA*2401, HLAB*5801, DRB1*0401, and DRB1*1501 [23,24,25]. The interaction of the epitope-HLA molecule was compared using the docking of the HLA protein with a peptide with which it was crystallized as a control. The peptides selected were those with lower interaction energy and affinity for the HLA in PyMOL visualization, suggesting a less forced interaction.

The dynamic behavior of the coupled molecules was evaluated to analyze the pattern of affinity and interaction since a strong interaction between the vaccine construct and the HLA receptors is necessary to elicit a stable immune response [33,50]. Interactions between the vaccine construct and HLA I and II proteins were identified; the results in the graphs and the RMSD showed that the designed vaccine construct has a trajectory like a receptor, which is consistent with the movement of the receptor, that is, they remain attached in the 50 ns of the simulation. This ability of the vaccine construct to interact with immune cell receptors indicates that it can trigger an adequate immune response against MTB.

A molecular docking approach was used to predict the binding of the designed vaccine construct with TLR4, MHC-I, and MHC-II molecules to understand the immune response towards the final vaccine structure. For all receptors, the vaccine acquired deep binding inside the pocket of the receptors, and recognition and binding between TLR-4 and RS09 adjuvant were also observed. This analysis was important to determine the presentation of the vaccine to the host immune system to activate immune signaling pathways and confer protective immunity [41,44].

In the vaccine design, the selected epitopes were separated by the GPGPG linker, and EAAAK linkers were added at the N-terminus of the construct [31,41,51]. RS09 adjuvant was also included, which is a bacterial LPS mimic that activates TLR4 [36]. RS09 is safe and more advanced than traditional Freund’s adjuvant, and it is needed to accelerate antigen-specific immunity, activate T and B cells, and increase the vaccine’s efficacy [44,47].

Because the PE_PGRS protein subfamily contains a high GC content (approximately 80%), its cloning and expression are difficult [52,53]; however, its ability to generate an immune response in the host cell is known [9,10,54]. Although there is little information about PE_PGRS49 and PE_PGRS56, two studies reported important characteristics that reinforce our findings and their potential as therapeutic targets. The first, an in-silico study on PE_PGRS49, predicted MHC-II binding sites as well as the presence of GTP-binding motifs, which might serve as a recognition target for human immune cells. PE_PGRS49 also exhibits motifs of an enzyme involved in amino acid biosynthesis and ligand-binding sites that are involved in amino acid and nucleotide biosynthesis, characteristics that suggest the participation of PE_PGRS49 in amino acid biosynthesis [55]. The second, a systematic study of the genetic determinants of TB transmission at the level of individual strains, was performed by molecular and conventional epidemiology with in vitro immunological assays to identify bacterial factors associated with tuberculosis transmission. Five markers of high TB transmissibility in vivo that are associated with altered immune responses in vitro were identified (PE_PGRS56, *aspE*, Rv0197, Rv2813–2814, and Rv2815–2816c). Of the five markers, only PE_PGRS56 affected T-cell cytokine responses [56].

It is important to note that the BCG vaccine strain does not secrete PE_PGRS proteins due to the deletion of the RD5 genetic region; this is likely to render BCG incapable of inducing immune responses against all PE_PGRS proteins [9]. There is evidence that a protein from this subfamily, PE_PGRS33, can trigger proinflammatory signals and promote phagocytosis of MTB by macrophages to control MTB replication [10]. In the search for a safe alternative vaccine that provides protection, it is proposed to use only some epitopes and not the complete proteins. In this way, they could obtain benefits such as avoiding the complications of cultivating the pathogen and reducing production costs [42,43,44,48]. Since it includes epitopes of two highly antigenic MTB proteins, this proposed vaccine candidate could elicit a protective immune response.

## 5. Conclusions

By using a reverse vaccinology strategy, a vaccine candidate comprising a molecular adjuvant and epitopes of the highly antigenic and exclusive PE_PGRS49 and PE_PGRS56 proteins was constructed. The results showed that epitopes GGAGGNGSLSS (128–138), FAGAGGQGGLGG (222–233), GIGGGTQSATGLG (274–286), and GTGWNGGKGDTG (1034–1045) are suitable candidates to be used in the development of a vaccine against TB and are expected to elicit humoral and cellular immune responses. Although in vivo and in vitro evaluation is still necessary to test their potential for protection against the disease, the bioinformatic analyses predict a stable and highly immunogenic construct.

## Figures and Tables

**Figure 1 microorganisms-11-01647-f001:**
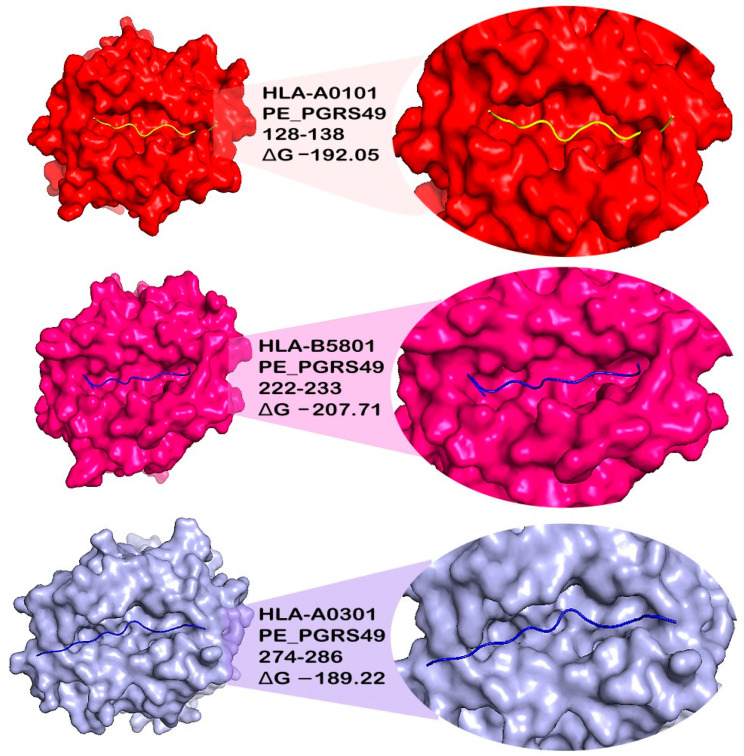
Selected epitopes docked to MHC alleles. The image shows a close-up of the MHC protein cavity to observe the docking between the PE_PGRS49 or PE_PGRS56 peptide and the MHC protein (surface representation). The epitopes (cartoon representation) cover the binding cleft, and the molecular docking energy value is indicated.

**Figure 2 microorganisms-11-01647-f002:**
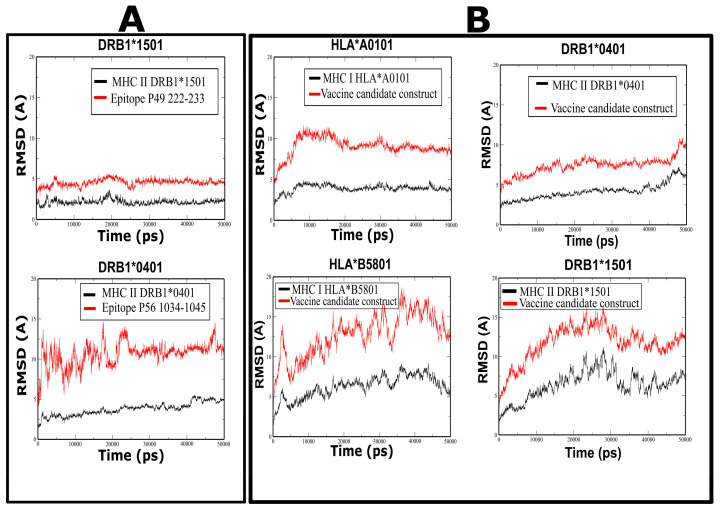
Stability of complexes assessed through RMSD values. (**A**) Stability of epitopes P49 and P56. (**B**) Stability of the vaccine construct complexed with HLA I (red) and II molecules (black). The RMSD stabilization in all cases indicates the stability of the complexes.

**Figure 3 microorganisms-11-01647-f003:**
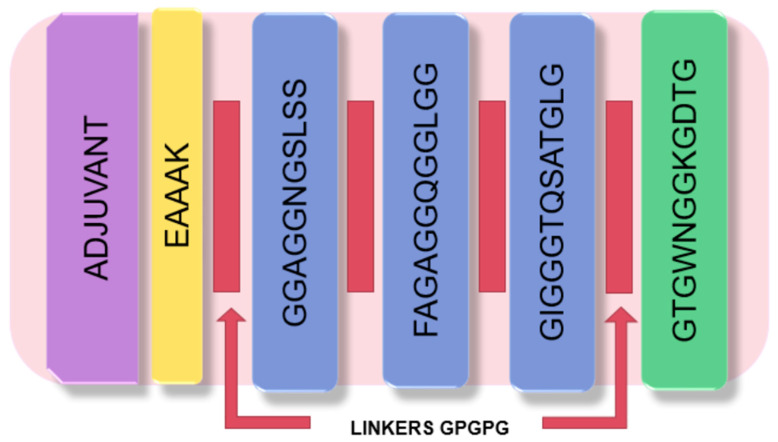
Schematic illustration of the vaccine construct. The vaccine construct is comprised of four epitopes: three from the PE_PGRS49 (blue) protein, one from the PE_PGRS56 (green) protein, and the adjuvant RS09 (purple). Additionally, the linkers EAAAK (yellow) and GPGPG (red) were added to the N-terminal and between epitopes, respectively.

**Figure 4 microorganisms-11-01647-f004:**
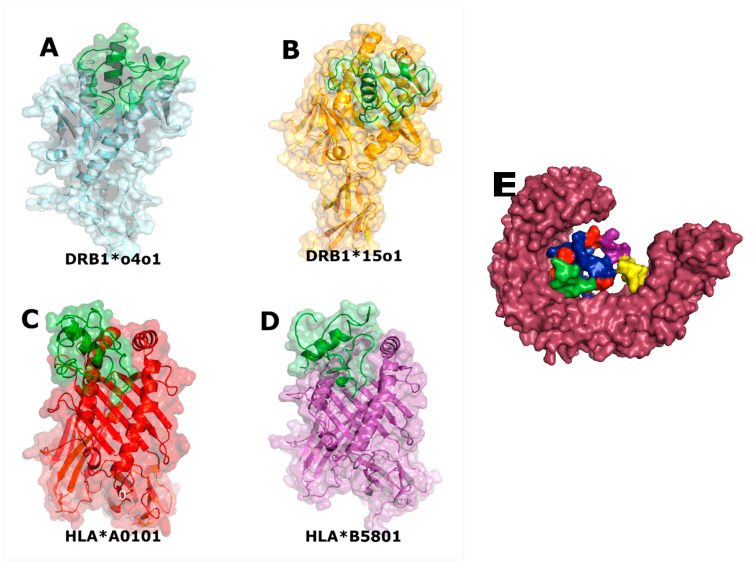
Molecular docking of the vaccine candidate construct–HLA and TLR-4. The image shows the behavior of the construct (green) with HLA I and II molecules (named in the image; (**A**–**D**)). Energy value (kcal/mol) of docked HLA–construct complexes: (**A**) −180.58; (**B**) −146.44; (**C**) −218.27; and (**D**) −175.7. (**E**) shows the construct complexed with TLR-4: adjuvant (yellow), linker EAAAK (purple), linkers GPGPG (red), PE_PGRS49 epitopes (blue), and PE_PGRS56 epitopes (green) with TLR-4 molecule. The energy value of the docked TLR-4-construct complex was −262.57 kcal/mol.

**Table 1 microorganisms-11-01647-t001:** Features of the selected epitopes.

PE_PGRS Protein	Epitope Sequence	Location	Size	Molecular Weight	Theoretical pI	Antigenicity	Allergenicity	Toxicity	Stability (Yes/No)	Estimated Half-Life (h)	Epitope Type
49	GSFGATSGPASIGVTG	58–73	16	1365.46	5.52	0.8794	Probable allergen	Not toxic	Yes	30	B-cell
	GSIGANSGIVGG	109–120	12	988.07	5.52	1.0119	Not allergen	Not toxic	Yes	30	B-cell, T-cell HLA I
	GGAGGNGSLSS	128–138	12	919.90	5.52	1.4603	Not allergen	Not toxic	Yes	30	B-cell, T-cell HLA I, II
	SGFFGGKGGFG	177–187	11	1017.11	8.47	0.0638	Probable allergen	Not toxic	Yes	1.9	B-cell, T-cell HLA I, II
	FAGAGGQGGLGG	222–233	12	948.00	5.52	2.2969	Not allergen	Not toxic	Yes	1.9	B-cell, T-cell HLA I, II
	GIGGGTQSATGLG	274–286	13	1075.14	5.52	2.0843	Not allergen	Not toxic	Yes	30	B-cell, T-cell HLA I, II
56	DAGKAGTGSAPGT	101–113	13	1089.13	5.84	1.8399	Not allergen	Not toxic	Yes	1.1	B-cell, T-cell HLA II
	GGAAGAATAG	338–347	10	702.72	5.52	2.0568	Probable allergen	Not toxic	Yes	30	B-cell, T-cell HLA II
	TVGGGTVPAGSGGQG	579–593	15	1201.26	5.19	2.1006	Not allergen	Not toxic	No	7.2	B-cell, T-cell HLA II
	NTANMTAQAG	780–789	10	978.04	5.52	0.9916	Probable allergen	Not toxic	Yes	1.4	B-cell, T-cell HLA II
	GTGWNGGKGDTG	1034–1045	12	1106.12	5.84	2.2536	Not allergen	Not toxic	Yes	30	T-cell HLA II

**Table 2 microorganisms-11-01647-t002:** Energy value (kcal/mol) of docked HLA–epitope complexes.

Protein	Epitope	HLA-A0101	HLA-A0201	HLA-A0301	HLA-A2401	HLA-B5801	DRB1*0401	DRB1*1501
PE_PGRS49	58–73	−211.89	−226.14	−225.61	−219.56	−229.65	NB	−214.04
	109–120	−195.58	−198.88	−191.89	−199.03	−198.8	NB	−160.73
	128–138	−192.05	−166.43	−181.6	−186.36	−182.14	NB	−163.5
	177–187	−220.21	−223.45	−240.43	−242.92	−262.45	NB	−221.46
	222–233	−191.82	−220.33	−207.86	−215.65	−207.71	−146.49	−183.23
	274–286	−190.24	−197.71	−189.22	−203.38	−199.63	NB	−165.28
PE_PGRS56	101–113	−193.22	−175.7	−170.51	−191.85	−188.23	NB	−156.42
	338–347	−175.35	−170.81	−162.55	−180.24	−176.91	NB	−136.44
	579–593	−203.23	−201.54	−182.91	−207.04	−197.08	−142.88	−179.87
	780–789	−206.4	−194.52	−214.13	−224.82	−205.7	NB	−167.75
	1034–1045	−201.1	−210.28	−216.18	−253.36	−209.21	−151.67	−197.28

NB = No binding to the HLA allele.

## Data Availability

Data will be made available on request.

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
