# Peer review of "Design of a Multi-Epitope Vaccine against Tuberculosis from Mycobacterium tuberculosis PE_PGRS49 and PE_PGRS56 Proteins by Reverse Vaccinology"

_microorganisms, 2023, doi:10.3390/microorganisms11071647_

Round 1
Reviewer 1 Report
An interesting paper addressing a growing yet long-standing concern
Queries:
The capability to generate immune responses by B and T cells.
How is this measured and what is measured? More information on types of responses/ tests as a limitation of measurements in studies (due to costs or technical availability) e.g. ELISA, antibodies etc
More information on the adjuvant RS09. What is it? An inert cassette or a viral vector.
Information seems to be garnered from pre-existing servers, such as the ‘VaxiJen server’. A brief description/ background of these servers and why they were chosen would help the reader.
Will this Vaccine also be effective for atypical mycoabcteria i.e. do they share the same targets?
Also given it promotes B and T cell immunity, is there a potential for use as TB treatment?
Author Response
Dear Editor, we appreciate the feedback provided by the revisors, which will greatly help us to improve the manuscript. Below you will find our response to the comments:
Reviewer 1. We appreciate all your comments and questions. They really helped us to improve the manuscript.
The capability to generate immune responses by B and T cells.
How is this measured and what is measured? More information on types of responses/ tests
as a limitation of measurements in studies (due to costs or technical availability) e.g. ELISA,
antibodies etc.
Response: Immunogenicity was inferred from predictions of B-cell epitopes, HLA and TLR binding, and antigenicity binding. We have modified Methodology section 2.7 to clarify the information. Regarding the limitations of in vitro assays in vaccine development, we included information in the Discussion section to address this topic.
More information on the adjuvant RS09. What is it? An inert cassette or a viral vector.
Response: Adjuvant RS09 is a peptide that mimics bacterial lipopolysaccharide and is used as an agonist of TLR4. RS09 has the following amino acid sequence: APPHALS. We have modified Methodology section 2.7 and the Discussion section to support the use of this adjuvant.
Information seems to be garnered from pre-existing servers, such as the ‘VaxiJen server’. A
brief description/ background of these servers and why they were chosen would help the
reader.
Response: We have modified the corresponding Methodology sections to include a brief description of each software/server used in the manuscript.
Will this Vaccine also be effective for atypical mycobacteria i.e. do they share the same
targets?
Response: Despite PE-PGRS proteins have been found in several mycobacterial species, proteins PE-PGRS49 and PE-PRGRS56 have been found only in Mycobacterium tuberculosis. Therefore, the proposed vaccine candidate would only be effective against M. tuberculosis. We performed an amino acid sequence alignment of all PE-PGRS49 and PE-PRGRS56 sequences found in the NCBI database and found that all the selected epitopes are conserved. We included this information in the Methodology section as well as supplementary information.
Also given it promotes B and T cell immunity, is there a potential for use as TB treatment?
Response: Potentially, the vaccine candidate proposed in our study could be of preventive or therapeutic use. In fact, there are some therapeutic vaccine candidates against tuberculosis that are being studied in clinical trials. According to other studies, these vaccines can be used in tree scenarios: they are administered to potentiate treatment during treatment of active disease, administered to prevent recurrence or relapse after standard treatment, or to prevent reactivation of latent tuberculosis to active tuberculosis through a Th1, or a mixed Th1/Th2 immune response, as well as potentially protective cytotoxic CD8 + T cells that could kill infected macrophages. We have included this information in the Discussion section.
I would also like to note that we had an error in the first epitope sequence, which was corrected to: GGAGGNGSLSS. This correction was also made in Figure 3.
Any modification made to the manuscript can be notice since we used the Track Changes in Word.
Please see the attachment

Reviewer 2 Report
The manuscript entitled “Design of a multi-epitope vaccine against tuberculosis from Mycobacterium tuberculosis PE_PGRS49 and PE_PGRS56 proteins by reverse vaccinology by Moreno et al. analyzed the two important antigenic mycobacterial proteins, PE_PGRS49 and PE_PGRS56 as a possible vaccine candidate using modern in silico methods. Further they selected the particular epitopes by considering several crucial properties including physicochemical, antigenicity, non-allergenic and non-toxicity.
Overall the study is well organized and explained. It will be helpful to screen some strong antigenic proteins and test as a vaccine or drug candidates.
I think this could be a great paper with some effort to enhance the introduction and discussion part by providing more references explaining the used methods and also about PE_PGRS49 and PE_PGRS56.
The authors need to show at least one in vitro experiment showing either cytokine profile or control of Mycobacterium tuberculosis in BMDMs or macrophage cell lines treated with predicted constructs.
Author Response
Dear Editor, we appreciate the feedback provided by the revisors, which will greatly help us to improve the manuscript. Below you will find our response to the comments:
Reviewer 2. We greatly appreciate your comments. They really helped us to improve the manuscript.
I think this could be a great paper with some effort to enhance the introduction and discussion part by providing more references explaining the used methods and, also about PE_PGRS49 and PE_PGRS56.
Response: We have included information about PE_PGRS49 and PE_PGRS56 in the Introduction section, we have included more information about the software used in this study, and we have enriched the Discussion section with the little information available for these proteins, which reinforces the great potential of PE_PGRS49 and PE_PGRS56 as a target for drug and vaccine development.
The authors need to show at least one in vitro experiment showing either cytokine profile or control of Mycobacterium tuberculosis in BMDMs or macrophage cell lines treated with predicted constructs.
Response: The aim of this study was to select epitopes with immunogenic potential against Mycobacterium tuberculosis, based on bioinformatics studies. This first step helped us to reduce costs during the in vitro evaluation by reducing the candidates, while assuring their efficacy. Further study of this candidate vaccine will include pre-clinical studies using in vitro cell culture assays as well as animal studies to corroborate the safety and efficacy of the vaccine. This information has been included in the Discussion section (lines 456-472).
I would also like to note that we had an error in the first epitope sequence, which was corrected to: GGAGGNGSLSS. This correction was also made in Figure 3.
Any modification made to the manuscript can be notice since we used the Track Changes in Word.
Please see the attachment